

# Oblique direction reach test: evaluating psychometric properties in stroke population

Rinita Mascarenhas[1], Akshatha Nayak[1], Abraham M. Joshua[1],
Shyam K. Krishnan[1], Vani Lakshmi R. Iyer[2], Jaya Shanker Tedla[3] and
Ravi Shankar Reddy[3]

[1] Department of Physiotherapy, Kasturba Medical College, Mangalore, Manipal Academy of Health Education, Manipal, India
[2] Department of Data Science, Prasanna School of Public Health, Manipal, Manipal Academy of Health Education, Manipal, India
[3] Department of Medical Rehabilitation Sciences, College of Applied Medical Science, King Khalid University, Abha, Saudi Arabia

## ABSTRACT

**Background:** Post-stroke individuals are observed to have reduced limits of stability (LOS) in all directions. Functional activities are rarely performed in pure cardinal planes; instead, they are most likely to be performed in an oblique direction. Existing tools are either expensive or sophisticated to assess the LOS in an oblique direction. Therefore, this study's primary objective is to evaluate the intra-rater, inter-rater reliability, and validity of the oblique direction reach test (ODRT) among stroke subjects.

**Materials & Methods:** A total of 96 first-time stroke patients with age, gender, height, and weight-matched healthy controls aged 18–80 years were recruited for the study. Oblique, forward, and lateral reach distances were assessed using the standard procedure of ODRT, Functional Reach Test (FRT), and Lateral Reach Test (LRT), respectively. Validity was tested by correlating the ODRT distance with the Berg Balance Scale (BBS) Score using Spearman's rank correlation coefficient. Intraclass correlation coefficients (ICCs) and Bland Altman analysis were used to establish inter-rater reliability. ICCs were used to find intra-rater reliability. The Mann-Whitney U test was used to establish the mean difference of the FRT, LRT, and ODRT. Spearman's rank correlation coefficient and linear regression were used to correlate the distance of FRT and LRT with ODRT.

**Results:** A high concurrent validity was found between BBS and ODRT with an $r$-value of 0.905 ($p < 0.001$). Inter-rater reliability was high with an ICC of 0.997 (95% CI [0.996–0.998]), and intra-rater reliability was highly significant with an ICC of 0.996 (95% CI [0.994–0.998]). The stroke subjects reached a significantly shorter distance than healthy individuals in FRT, ODRT, and LRT. ODRT was highly correlated with FRT ($r = 0.985$) and LRT ($r = 0.978$) ($p < 0.001$) and had an $R^2 = 0.987$.

**Conclusion:** ODRT is a highly valid and reliable tool that can be used to evaluate balance in stroke patients. Individuals who reached less in the forward and lateral directions showed reduced reach distance in the oblique direction.

Corresponding author
Akshatha Nayak,
akshatha.nd@manipal.edu

# INTRODUCTION

Globally, stroke is the second leading cause of mortality and morbidity (*Feigin et al., 2021*). The crude prevalence of stroke in India is 26 to 757 out of 100,000 people per year (*Jones et al., 2022*). Post-stroke, individuals often develop sensory-motor, cognition, and balance impairments, leading to significant disability over time (*Wolfe, 2000*; *Gadidi et al., 2011*).

Balance is considered essential, as it contributes to walking and standing abilities (*Khan & Chevidikunnan, 2021*). Individuals who have experienced a stroke may have a balance deficit due to muscle weakness, incoordination, and the presence of neurological lesions (*Karthikbabu et al., 2012*). Poor balance is responsible for fall incidence as high as 40% within the first year following a stroke (*Samuelsson, Hansson & Persson, 2019*). Participation in daily activities among the stroke population is affected by the high risk of falls, implying a low quality of life (*Schmid et al., 2013*).

Therefore, it is essential to assess balance among stroke patients (*Lin et al., 2004*). Dynamic posturography is the existing gold standard measure used to assess balance (*Lin et al., 2004*). However, the application of this instrument is not feasible because the equipment needed is expensive and sophisticated. However, the Berg Balance Scale (BBS), the Time Up and Go (TUG), the Tinetti POMA scale, the Multidirection Reach Test (MDRT), the Lateral Reach Test (LRT), and the Functional Reach Test (FRT) are additional clinical measures that assess balance (*Lin et al., 2004*; *Blum & Korner-Bitensky, 2008*; *Canbek et al., 2013*; *Hwang et al., 2015*). Among these tests, LRT, MDRT, and FRT are the most feasible tests because they can be done quickly and do not require any use of any complex or expensive equipment (*Duncan et al., 1990*; *Newton, 2001*; *Katz-Leurer et al., 2009*; *Deshmukh, Ganesan & Tedla, 2011*; *Hwang et al., 2015*). FRT and LRT assess an individual's potential to reach ahead or sideways, causing their center of mass (COM) to shift over a firm base of support (BOS), and assessing their ability to control their body (*Duncan et al., 1990*; *Newton, 2001*; *Katz-Leurer et al., 2009*; *Deshmukh, Ganesan & Tedla, 2011*; *Verheyden et al., 2011*). Additionally, MDRT examines an individual ability to displace their COM in four ways: forward, backward, right, and left (*Hwang et al., 2015*).

However, in most daily tasks, reaching mainly occurs in the oblique plane rather than forward or sideways (*Van Andel et al., 2008*). During reaching activities, the center of pressure (COP) is displaced to the opposite side of the arm used for reaching (*Tessem, Hagstrøm & Fallang, 2007*). This situation may cause falls due to an individual's inability to voluntarily move their body within LOS (*Nachreiner et al., 2007*; *Van Andel et al., 2008*). *Van Dijk et al. (2017)* reported that COP excursion of the affected side in the oblique, and posterior directions was challenging. Furthermore, the COP distances were negatively correlated with the fear of falling among stroke survivors (*Van Dijk et al., 2017*). Lack of coordination between pelvic stabilization muscles that contributes to poor muscle recruitment among recovering stroke patients could lead to instability while performing activities of daily living (ADLs) (*Goldie et al., 1996*; *Van Dijk et al., 2017*; *Park, 2018*).

The abovementioned tests are designed to assess limits of stability (LOS) in forward, lateral, or backward direction and do not evaluate balance in an oblique direction. Therefore, the oblique direction reach test (ODRT) has the potential to serve as one of the tests to assess certain activities of daily living (ADLs), such as reaching activities among stroke survivors. ODRT requires as individual to firmly place their foot on the floor and reach in the oblique plane, without any extraneous movements, such as flexing their knees or taking a step forward (*Tedla et al., 2020*, *2021*). Alongside assessing dynamic balance, it indirectly measures the LOS (*Tedla et al., 2020*, *2021*). The test has previously been used to establish normative data in children and young people and has excellent psychometric properties (*Tedla et al., 2019*, *2020*).

However, there is a near total absence of retrievable literature on the application of ODRT on stroke subjects. Therefore, this study aimed to establish the psychometric properties of ODRT among the stroke population and to evaluate whether ODRT can be feasibly used to evaluate LOS among individuals who have experienced a stroke. Furthermore, it examines whether there is discrepancy in reach distance for oblique direction between individuals who have had a stroke and healthy individuals.

## MATERIALS AND METHODS

### Study design

An analytical cross-sectional study with 192 participants (96 post-stroke individuals, 96 healthy controls) were included to evaluate the psychometric properties of ODRT. The psychometric properties assessed were intra-rater reliability, inter-rater reliability, and validity. Ethical approval for the study was obtained from the Institutional Ethics Committee (IEC), Kasturba Medical College, Manipal Academy of Higher Education, Mangalore, India (IEC KMC MLR 01/2022/11). Before recruiting the participants, the trial was registered in the Clinical Trial Registry of India (CTRI/2022/05/042353).

### Subjects

Ninety-six participants diagnosed with a single episode of stroke were included in the study. All participants had been admitted to Kasturba Medical College Hospital, Mangalore. Participants that were clinically stable, aged 18–80, scored >26 cognitively on the Montreal Cognitive Assessment (MoCA), and could stand independently for 2 min were included. Participants were excluded if they had an underlying neurological condition other than stroke. Participants with musculoskeletal and cardiovascular conditions that may have affected the study outcome were also excluded. Ninety-six healthy participants were recruited based on the recruited stroke individual's age, gender, height, and weight. The healthy subjects were excluded if they had any underlying neurological, musculoskeletal, or cardiovascular conditions that may have affected the study outcome. All the participants were informed of the procedure, and a written informed consent was obtained from all participants before recruitment.

## Outcome variables

### Oblique direction reach test

Oblique direction reach distance which was measured by ODRT. ODRT has been reported to have intra-rater and inter-rater reliability among young-adults, with intra-class correlation coefficient values of 0.97 and 0.86. Concurrent validity with the forward and lateral reach tests was found to have $r$-values of 0.78 and 0.73, (*Tedla et al., 2020*). The authors have permission to use this instrument from all the copyright holders.

### Berg balance scale

BBS has 14 items, which use direct observation to quantitatively assess balance and the risk of falls after a stroke. Studies have shown that the scale's internal consistency is excellent (Cronbach alpha 0.92–0.98). Furthermore, the ICC value for inter-rater reliability was 0.98; for intra-rater reliability was 0.97, and test-retest reliability was 0.98. It also scored highly on other measurement scales (*Berg, 1989*; *Blum & Korner-Bitensky, 2008*). The authors obtained permission from the copyright holders to use this instrument.

### Functional reach test

FRT assesses the LOS by measuring the maximum distance an individual can reach forward while standing in a fixed position. FRT's inter-reliability was found to be 0.987. Intra-rater reliability was found to be 0.983 (*Duncan et al., 1990*).

### Lateral reach test

LRT assesses the LOS by measuring the maximum distance an individual can reach laterally while standing in a fixed position. The ICC value of intra-rater reliability was 0.92 for the lateral reach test scores (*Tedla et al., 2019*).

## Procedure

Information about the participants' age, gender, height, weight, affected side, dominant side, and duration of stroke was noted.

A wooden ruler was affixed to a tripod stand that was placed at the level of the subject's acromion process. The wooden ruler was placed at 45 degrees between the anterior and lateral directions. The reference for 45 degrees was measured on the floor by marking it with tape. Then, the participant was instructed to stand with their feet shoulder-width apart and their unaffected arm raised to 90 degrees obliquely in flexion and horizontal abduction with the hand in a fist. The scale was placed behind the participant's arm. The participant was then instructed to hold this position, and the researcher noted this as the starting point against the ruler. The level of the third metacarpal head of the fist was taken as a reference (Fig. 1).

The participant was then instructed to reach obliquely as far as possible without flexing their knees, taking a step, or being supported by the stand. Displacement of the third metacarpal head at the end of reaching was noted. The difference in distance between the starting and ending points was noted as the oblique reach distance (Fig. 1).

Two independent researchers performed the procedure on the same occasion, and the values generated were used to explore inter-rater reliability. The same researcher obtained

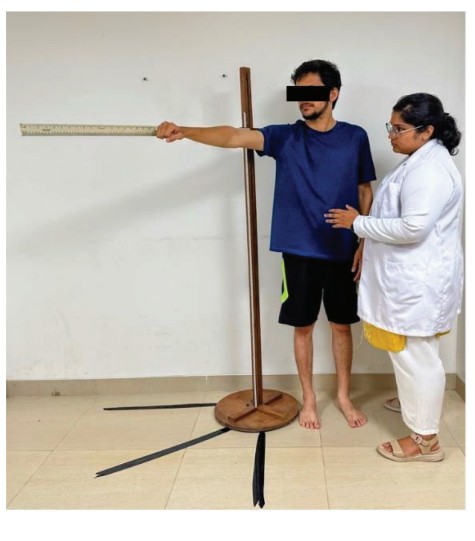
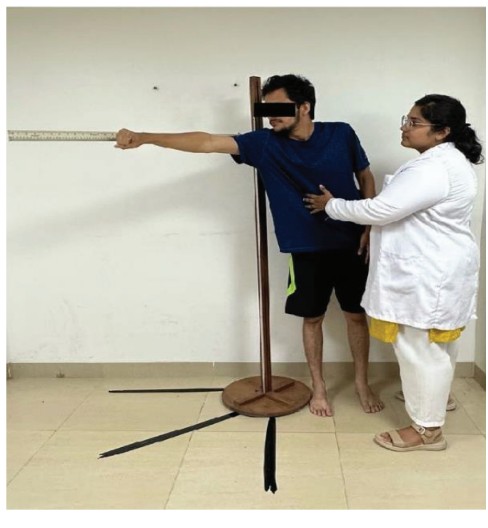

(a)                                                    (b)

**Figure 1  Assessment of the oblique direction reach test: (A) and (B).**

measures of the ODRT within a washout period of 2 h to assess the intra-rater reliability. BBS and ODRT were correlated to establish concurrent validity. The standard procedure was followed while assessing the FRT and the LRT among participants.

## Data analysis

Jamovi 2.3.21 was used for the analysis process. The mean and standard deviation (SD) of age, height, and weight were expressed using descriptive statistics. The Shapiro-Wilk test was used to evaluate the normal distribution and it was found that the data was non-normally distributed. The concurrent validity of ODRT with respect to BBS was analyzed by estimating Spearman's rank correlation. ICC and Bland-Altman plots were estimated to establish inter-rater and intra-rater reliability. Spearman's rank correlation was used to co-relate ODRT with FRT and LRT. The Mann-Whitney U test was applied to test whether there is a significant mean difference between the case and control groups while performing ODRT, FRT, and LRT. Linear regression was used to explore the relationship between ODRT values and FRT, and LRT values.

## RESULTS

A total of 192 individuals were recruited for the study. The mean age was found to be 61.97 SD years. The descriptive data of the study participants are in Table 1.

### Psychometric properties

Correlating the scores of ODRT with BBS showed a high correlation. The $r$ value was 0.905 ($p < 0.001$), indicating a high validity for the ODRT (Table 2). The inter-rater reliability was high, with an ICC of 0.997(95% CI [0.996–0.998]) ($p < 0.001$). The intra-rater reliability analysis showed a strong ICC of 0.996 (95% CI [0.994–0.998]) ($p < 0.001$).

**Table 1 Descriptive data of the stroke subjects.**

| Variables | | Mean ± SD | |
|---|---|---|---|
| Age (years) | | 61.917 ± 10.695 | |
| Height (cm) | | 164.453 ± 10.028 | |
| Weight (kg) | | 63.365 ± 10.271 | |
| BMI | | 23.363 ± 2.810 | |
| Post-stroke duration (days) | | Median (IQR) | |
| | | 5 (5.0, 10.0) | |
| | | *n* | % |
| Gender | Male | 58 | 60.5% |
| | Female | 38 | 539.5% |
| Affected side | Right | 60 | 62.5% |
| | Left | 36 | 37.5% |

**Table 2 Concurrent validity between oblique direction reach test and Berg Balance Scale.**

| | Median (IQR) | *r* | *p*-value |
|---|---|---|---|
| ODRT (cm) | 12.3 (8.00, 16.6) | 0.905 | <0.001* |
| BBS | 48 (40.8, 49.0) | | |

**Note:**
* Significant

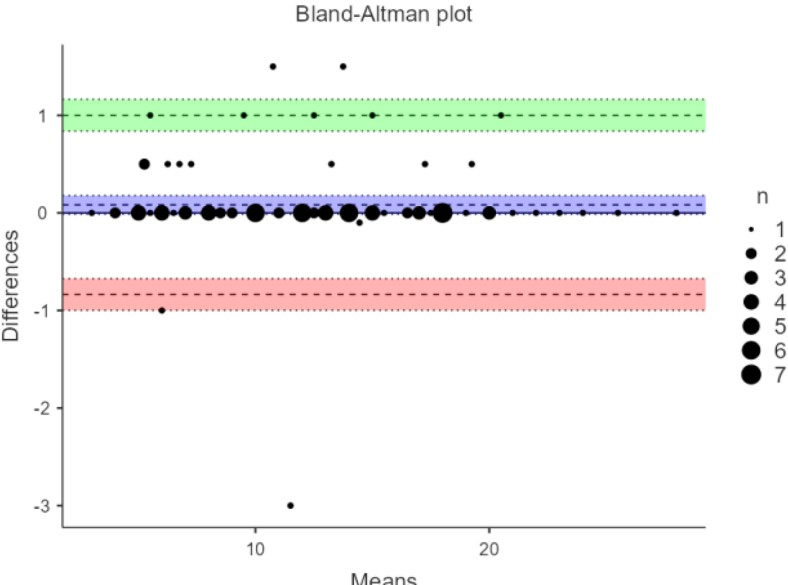

**Figure 2 Bland-Altman plot for inter-rater reliability of oblique direction reach test among stroke subjects.**

The results of inter-rater and intra-rater reliability are shown in the Bland-Altman plot analysis (Figs. 2 and 3).

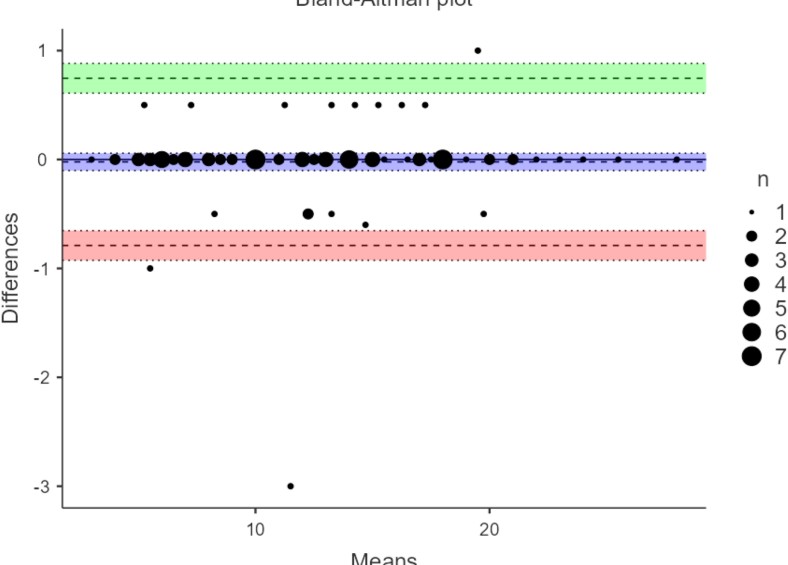

**Figure 3  Bland-Altman plot for intra-rater reliability of oblique direction reach test.**

**Table 3  Descriptive values of oblique direction reach test, Forward Reach Test, and Lateral Reach Test.**

|  | Type | Median (IQR) | *p-value* |
|---|---|---|---|
| ODRT (cm) | Case | 12.3 (8.0, 16.6) | <0.001[*] |
|  | Control | 20.0 (19.0, 25.0) |  |
| FRT (cm) | Case | 15.0 (10.0, 19.0) | <0.001[*] |
|  | Control | 23.0 (21.5, 28.0) |  |
| LRT (cm) | Case | 10.0 (5.8, 14.6) | <0.001[*] |
|  | Control | 18.0 (16.8, 22.1) |  |

Note:
[*] Significant.

**Table 4  Correlation of oblique direction reach test with Forward Reach Test & Lateral Reach Test.**

|  | Variables | *r* | *p*-value |
|---|---|---|---|
| ODRT | FRT | 0.985 | <0.001[*] |
|  | LRT | 0.978 | <0.001[*] |

Note:
[*] Significant.

**Table 5  Regression analysis of oblique direction reach test with Lateral Reach Test & Forward Reach Test.**

| Predictor | Estimate | SE of estimate | *t* | *p*-value | $R^2$ |
|---|---|---|---|---|---|
| Intercept | −0.269 | 0.204 | −1.32 | 0.188 | 0.987 |
| LRT | 0.556 | 0.035 | 16.07 | <0.001[*] |  |
| FRT | 0.423 | 0.036 | 11.64 | <0.001[*] |  |

Note:
[*] Significant

On average subjects without stroke were able to reach 9.50 and 9.00 cm further in FRT and LRT, respectively (Table 3).

A high correlation was observed between ODRT, FRT, and LRT with a high significance ($p < 0.001$) and $R^2 = 0.987$. The correlations and regression analysis are provided in Tables 4 and 5.

## DISCUSSION

The study's primary aim was to determine the psychometric properties of ODRT among individuals who had experienced a stroke. The psychometric properties evaluated in this study were inter-rater reliability, intra-rater reliability, and concurrent validity. ODRT measures one's ability to displace their center of gravity away from the base of support in an oblique direction without taking a step or receiving any support. The study's secondary aim was to compare the ODRT of individuals who experienced stroke with age, BMI, and gender-matched controls, as well as to evaluate if there was a correlation between the performance in ODRT and FRT & LRT in subjects with and without stroke.

Excellent correlation has been previously identified with BBS and other scales of balance impairment (Blum & Korner-Bitensky, 2008). The performances obtained in ODRT were correlated with BBS scores to evaluate the concurrent validity of ODRT. A high correlation was observed between BBS and ODRT among stroke subjects, with an r-value of 0.905.

This study's results demonstrate that ODRT has good intra-rater reliability when used to assess LOS in individuals diagnosed with stroke. Moreover, the Bland-Altman plot showed no bias in the measurements made by the same researcher; most of the values came within the upper and lower limits of mean differences.

The inter-rater reliability of ODRT has not been previously established in any adult population to the best of our knowledge. The current study found high inter-rater reliability for ODRT when applied in subjects with stroke. Therefore, the tool is consistent among raters.

The study identified that the forward, oblique, and lateral reach distances of participants who had experienced strokes were lower than those of healthy individuals. Similar, results were reported by Merchán-Baeza, González-Sánchez & Cuesta-Vargas (2015) who found that stroke survivors exhibited reduced functional reach. Post-stroke, a deterioration occurs in the function of trunk muscles on both sides of the body, affecting the proximal control. The lack of proximal stabilization significantly impacts the upper limb function; this could attribute to poor reaching distance within their LOS among individuals who have experienced a stroke (Karthikbabu et al., 2012).

Participants who had experienced strokes attained lower ODRT scores, and there was also a reduction in the distances reached in FRT and LRT. Similarly, Tedla et al. (2021) found that as the ODRT distance increased, there was an increase in the distance obtained in FRT. This finding indicates that there could be an association between reaching abilities in cardinal as well as oblique directions. This may be due to impairment in trunk muscle strength in post-stroke patients (Karthikbabu et al., 2012). This study identified a high correlation between the distances reached in ODRT, and FRT & LRT distances.

The strength of our study is a fairly high sample size which is good enough to authoritatively establish validity and reliability of a test. We found ODRT to be an inexpensive and time-efficient tool that can assess the LOS of a post-stroke individual.

This study has limitations because the concurrent validity of ODRT was not tested against more accurate research-oriented tools to evaluate balance, such as force platforms. Furthermore, stroke individuals were not classified according to the lesion's site and the stroke's duration in the inclusion criteria. The site and duration of the stroke may have impacted the individuals' reach distance.

Future researchers may test the criterion-based validity of ODRT against more objective measures, such as the COP excursion, in participants with and without stroke. Furthermore, researchers should evaluate the correlation of ODRT distance with trunk muscle strength and recruitment pattern using EMG analysis in post-stroke patients.

## CONCLUSIONS

This study has identified that ODRT has excellent inter-rater, intra-rater reliability, and validity in assessing oblique reach distance among stroke subjects. ODRT can be used among stroke subjects to assess the limits of stability and functional balance because it is easy and inexpensive. Notably, individuals who reached less distance in the oblique direction had shorter reach distances in the forward and lateral directions.

## ACKNOWLEDGEMENTS

The authors would like to express their gratitude to all the participants in this study.

### Funding

This work was supported by the Deanship of Scientific Research at King Khalid University through large group Research Project under grant number RGP2/328/44. The funders had no role in study design, data collection and analysis, decision to publish, or preparation of the manuscript.

### Grant Disclosures

The following grant information was disclosed by the authors:
Deanship of Scientific Research at King Khalid University: RGP2/328/44.

### Competing Interests

The authors declare that they have no competing interests.

### Author Contributions

- Rinita Mascarenhas conceived and designed the experiments, performed the experiments, prepared figures and/or tables, and approved the final draft.
- Akshatha Nayak conceived and designed the experiments, performed the experiments, authored or reviewed drafts of the article, and approved the final draft.

- Abraham M. Joshua conceived and designed the experiments, authored or reviewed drafts of the article, and approved the final draft.
- Shyam K. Krishnan analyzed the data, authored or reviewed drafts of the article, and approved the final draft.
- Vani Lakshmi R. Iyer analyzed the data, prepared figures and/or tables, and approved the final draft.
- Jaya Shanker Tedla conceived and designed the experiments, prepared figures and/or tables, and approved the final draft.
- Ravi Shankar Reddy performed the experiments, authored or reviewed drafts of the article, and approved the final draft.

## Human Ethics

The following information was supplied relating to ethical approvals (*i.e.*, approving body and any reference numbers):

The Ethics approval was granted by the Institution Ethics Committee, Kasturba Medical College, Mangalore.

Protocol No.: IEC KMC MLR 01/2022/11

## Data Availability

The raw measurements are available in the Supplemental File.

## Supplemental Information

Supplemental information for this article can be found online at http://dx.doi.org/10.7717/peerj.16562#supplemental-information.

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
