# Peer review of "Oblique direction reach test: evaluating psychometric properties in stroke population"

_PeerJ, doi:10.7717/peerj.16562_

## Round 0.1 · original submission · Major Revisions

Please improve the spelling and syntax errors. All reviewers think the introduction can be improved to better set up the basis for this work.

**Language Note:** The Academic Editor has identified that the English language must be improved. PeerJ can provide language editing services - please contact us at copyediting@peerj.com for pricing (be sure to provide your manuscript number and title). Alternatively, you should make your own arrangements to improve the language quality and provide details in your response letter. – PeerJ Staff

Reviewer 1 ·

Basic reporting

This study investigated possibility of ODRT as a feasible tool to estimate limits of stability of stroke patients. I think that the topic is very interesting with meaningful findings. The authors can increase the clarity of their findings with reflecting some minor suggestions.

1. In the Introduction, more specific information on ODRT (e.g., test components, methods, and rationale) need to be provided (line 87-90). Why is ODRT suitable for assessing the stability of patients with stroke?
2. In line 99, may be typo (i.e., 96 age, ?)
3. In the Results, first paragraph (lines 181-184) should be moved to the Methods (subject)
4. In the Methods, please list specific variables from each test in Data Analysis.

Experimental design

No comment

Validity of the findings

No comment

Additional comments

No comment

Reviewer 2 ·

Basic reporting

More background about the research needs should be provided
Line 81-82 ‘reaching is mainly in the oblique plane in comparison to forward or sideways’. This statement is quite important as it forms the research need of the study. Please elaborate on this statement with findings from the literature.

Experimental design

The research question is about investigating the oblique reaching ability of stroke patient, it seems that the study investigates the oblique reach to the unaffected size only, which does not make sense to me as there should be significant differences between reach to affect and unaffected size.
Methods is not clearly described
How were the patients being recruited?
Line 183 ‘anthropometric-matched controls’ how do you match the stroke patient and the health control? And what parameters have been used for the matching ?

Validity of the findings

Can you compare the ODRT distance between health control and that reported in the previous study to illustrate that the results is comparable to the previous study?

Reviewer 3 ·

Basic reporting

The writing needs to be improved.
The introduction has many paragraphs, even though the main ideas are the same. It can be improved in 3 or more main paragraphs following 3 main ideas: 1°stroke &disability &balance impairments; balance assessment tools and their advantages and limitations; 3°the novelty of ODRT.

Spelling mistakes, syntax errors and long sentences can be corrected: Line 20 "observed"; line 27 "....patients and ? age": proofreading and linguistic correction by a professional would be a good touch

It would be important to explain why this test is more important than the MDRT " Hwang WJ, Kim JH, Jeon SH, Chung Y. Maximal lateral reaching distance on the affected side using the multi-directional reach test in persons with stroke. J Phys Ther Sci. 2015 Sep;27(9):2713-5. doi: 10.1589/jpts.27.2713. Epub 2015 Sep 30. PMID: 26504275; PMCID: PMC4616076."

There are lots of tables (7 tables) . Some tables can be merged (Bland Altman) or deleted for presentation in the text (correlation).

The discussion can be improved :avoid repeating data and focus on interpretations and criticisms (lines 211, 215°

Experimental design

no comment

Validity of the findings

Although the assessment of oblique direction using easy and less costly tools is very interesting, the novelty of this study can be questioned because nowhere in the introduction or discussion the author confronts the advantage of the ODRT to the multi-directional reach test. "Hwang WJ, Kim JH, Jeon SH, Chung Y. Maximal lateral reaching distance on the affected side using the multi-directional reach test in persons with stroke. J Phys Ther Sci. 2015 Sep;27(9):2713-5. doi: 10.1589/jpts.27.2713. Epub 2015 Sep 30. PMID: 26504275; PMCID: PMC4616076."

The conclusion of the abstract is different from the general conclusion. I think that the first sentence in the abstract: Lines 45-46 does not address any of the objectives of this study. It could be moved to the end of the conclusion

Additional comments

It is interesting to assess balance in the oblique direction in stroke victims with easy and low-cost tools, this paper adresses an interesting topic.
The statistical methods used and the sample size make the paper interesting.

Major comment:
1. " the novelty of the ODRT compared to the multi-directional reach test (MDRT) need to be well highlighted in the introduction"
2. A linguistic revision and improved syntax would make the paper more readable (long introduction with same ideas, long sentences sometimes with lack of clarity, e.g line 61; line 63 " limited instead of affected"; lines 95_96; line 61; lines 116_117; ..
3. Number of tables can be shorten

see the annotated paper

Annotated reviews are not available for download in order to protect the identity of reviewers who chose to remain anonymous.

---

## Round 0.2 · accepted · Accept

Thank you for responding to reviewer comments and improving the manuscript.

Reviewer 1 ·

Basic reporting

No further issue exists.

Experimental design

no comment

Validity of the findings

no comment

Additional comments

no comment

Reviewer 3 ·

Basic reporting

The paper is now much improved. Many of my recent comments have been taken into account. The introduction seems clear with few but important references. Figures and table are well designed

Experimental design

Research question and methodology are now clear

Validity of the findings

oblique direction, although not always essential in everyday life, is a key element in post-stroke balance. ODRT could therefore make an enormous contribution to the assessment of stroke victims and the planning of treatment according to the needs of each patient.

Additional comments

No comment